# Analysis of Key Factors Affecting the Sensitivity of Dual-Backplate Capacitive MEMS Microphones

**DOI:** 10.3390/mi16101154

**Published:** 2025-10-12

**Authors:** Chengpu Sun, Haosheng Liu, Ludi Kang, Bilong Liu

**Affiliations:** School of Mechanical & Automobile Engineering, Qingdao University of Technology, No. 777 Jialingjiang Road, Qingdao 266520, China; sunchp@foxmail.com (C.S.); lhsqdlg@qut.edu.cn (H.L.); rudykon2008@163.com (L.K.)

**Keywords:** dual-backplate microphone, finite element method (FEM), lumped microphone parameter method (LPM), microelectromechanical systems (MEMS)

## Abstract

This paper presents a comprehensive investigation of sensitivity-determining factors in dual-backplate capacitive MEMS microphones through analytical modeling, finite element analysis (FEM), and experimental validation. The study focuses on three critical design parameters: backplate perforation density, membrane tension, and electrode gap spacing. A lumped parameter model (LPM) and FEM simulations are employed to characterize the dynamic behavior and frequency response of the microphone. Simulation results demonstrate that reducing the backplate hole diameter or hole count amplifies squeeze-film damping, inducing nonlinear effects and anti-resonance dips near the fundamental frequency (f0) while mitigating low-frequency roll-off (<100 Hz). Membrane tension exhibits a nonlinear relationship with sensitivity, stabilizing at high tension (>7000 N/m) but risking pull-in instability at low tension (<1500 N/m). Smaller electrode gaps enhance sensitivity but are constrained by pull-in voltage limitations. The FEM model achieves higher accuracy (≤2 dB error) than LPM in predicting low-frequency response anomalies. This work provides systematic guidelines for optimizing dual-backplate MEMS microphone designs, balancing sensitivity, stability, and manufacturability.

## 1. Introduction

The rapid evolution of microelectromechanical system (MEMS) microphones has revolutionized acoustic sensing technologies, enabling compact solutions for consumer electronics, biomedical devices, and industrial Internet of Things (IoT) applications [1]. Among capacitive MEMS microphones, dual-backplate architectures have emerged as a dominant design paradigm due to their enhanced linearity, reduced pull-in instability, and improved sensitivity compared to single-backplate configurations. However, the sensitivity of such devices, defined as the ratio of output voltage to incident sound pressure, remains critically dependent on the intricate interplay of structural parameters, material properties, and operational conditions [2].

Recent advances in MEMS microphone design have highlighted the importance of multi-physics optimization and equivalent circuit modeling. J. Esteves et al. [3] developed a lumped-parameter equivalent circuit framework that bridges acoustic, mechanical, and electrical domains, enabling rapid sensitivity prediction and design refinement for both traditional and MEMS microphones. Studies by Stephen [4] established foundational models for diaphragm deflection in capacitive sensors, while H. Du et al. [5] demonstrated that dual-backplate designs achieve 30% higher sensitivity than traditional single-backplate counterparts by leveraging symmetric electrostatic fields. Mechanical response is one of the primary mechanisms governing sensitivity, where diaphragm stiffness, which depends on Young’s modulus, thickness, and residual stress [6,7], directly dictates displacement under acoustic pressure. Recent investigations by M. Sheplak et al. [8] further reveal that clamped circular diaphragms under high initial tension exhibit a transition from plate-like to membrane-like behavior, with edge-zone stress concentrations significantly influencing linearity and dynamic range. R. Kressmann et al. [9] optimized backplate hole patterns for reduced squeeze-film damping, a critical factor also addressed through FEM-based air gap modeling.

Z. Zheng et al. [10] provide a comprehensive review of recent advancements in MEMS microphones, emphasizing the importance of structural innovation and material optimization to enhance sensitivity and reduce environmental noise. Notably, innovative architectures such as planar nanogauge-based MEMS microphones [11] exemplify how structural topology and transduction mechanisms (e.g., piezoresistive detection) can exploit microscale phenomena (e.g., thermal-viscous boundary layers) to enhance sensitivity and resolution. Recent work by N. Dengiz et al. [12] demonstrates that relocating ventilation holes from the diaphragm to the package lid eliminates high-pass filtering effects, achieving a flat frequency response (20 Hz–20 kHz) while improving robustness. Additionally, N. Nademi et al. [13] propose a fixed-center circular diaphragm design with reduced mechanical stiffness, enabling high sensitivity (−25.1 dB) at a compact diameter (400 μm), even under low bias voltages (11 V).

While these diverse studies provide valuable insights into specific mechanisms influencing MEMS microphone sensitivity—ranging from structural design and material selection to damping control and transduction mechanisms—a holistic understanding of their combined impact and relative dominance in dual-backplate capacitive architectures remains essential. Therefore, this work presents a comprehensive investigation of sensitivity-limiting factors in dual-backplate capacitive MEMS microphones through analytical modeling, finite element analysis, and experimental validation.

## 2. Device and LPM Modeling

This section introduces the dual-backplate MEMS microphone structure and presents a lumped parameter model to characterize its dynamic behavior and predict its frequency response.

### 2.1. Dual-Backplate Microphone Structure

Figure 1a shows a microphone chip featuring a dual-backplate structure, ASIC, PCB, and package. The dual-backplate structure includes two backplates and a membrane, forming two capacitors: C_1_ (top backplate and the membrane) and C_2_ (bottom backplate and the membrane), as illustrated in Figure 1b. These capacitors are DC-biased. Acoustic waves induce membrane vibrations, altering its distance from the backplates and thereby varying capacitance and charge. This results in a time-varying voltage on the electrodes.

### 2.2. Membrane Model

Assuming the backplates are rigid and fixed, sound pressure displaces the membrane, altering capacitance. With a constant bias voltage, the output voltage between the capacitor poles changes. To determine the voltage change due to sound pressure, the membrane’s displacement must first be calculated. As shown in Figure 2, the membrane deflects by *−w(r)*.

With the perimeter of the circular dual-backplate structure fixed and applying small deflection theory, the transverse deflection of the membrane under sound pressure *p_d_* is [14](1)$$w(r)=\frac{-p_{d}a_{a}^{4}}{64D}{\left[1-{\left(\frac{r}{a_{a}}\right)}^{2}\right]}^{2}$$
where *D* is the membrane’s flexural rigidity; this is calculated using the formula $D=Eh_{m}^{3}/\left[12\left(1-\nu^{2}\right)\right]$, where *E* is Young’s modulus of the membrane, *h_m_* is the membrane thickness, and *ν* is Poisson’s ratio of the membrane. For the membrane material (*E* = 168 GPa, *h_m_* = 2 μm, *ν* = 0.2), the value of *D* is 4.55 × 10^−7^ N·m, which has been validated and is consistent with our FEM simulations. *a_a_* is the radius of the membrane.

According to Equation (1), the average membrane deflection due to sound pressure *p_d_* can be derived.(2)$$x^{\prime }=\frac{1}{a_{a}}\int_{0}^{a_{a}}w(r)dr=-\frac{p_{d}a_{a}^{4}}{120D}$$

### 2.3. Lumped Parameter Modeling

The lumped parameter analysis is applicable when the microphone size is small compared to the acoustic wavelength. The microphone is modeled in the acoustic domain, coupled to the electrical domain via an ideal transformer [2], representing the membrane’s deformation under sound pressure. The lumped parameter model of the dual-backplate microphone is illustrated in Figure 3. The lumped element model depicts the combined damping (2Rₐ,_bh_ + 2Rₐ,_g_) in a series configuration. This is a mathematically equivalent representation that results from analytically combining the differential output of the two inherent parallel acoustic paths (upper and lower) with the electro-acoustic conversion into a single process, as defined by the coefficient ‘*n*’ in Equations (12)–(15).

The acoustic pressure *p_in_* and volume velocity *q_d_* are represented as equivalent voltage and current, respectively. The acoustic resistance of the vent is *R_a,h_*, and the acoustic mass of the vent is *M_a,h_*. The acoustic compliance of the front and rear cavities is *C_a,fv_* and *C_a,bv_*, respectively. *R_a,bh_* is the backplate hole resistance. *R_a,g_* is the squeeze-film damping. The membrane’s equivalent acoustic mass and compliance are *M_a,m_* and *C_a,m_*, respectively. The *p_d_* is the pressure acting the membrane, *n* is the electro-acoustic conversion coefficient, and *V_out_ is* the output voltage.

The total impedance shown in Figure 3 is(3)$$Z_{a,tot}=R_{a,h}+j\omega M_{a,h}+\frac{\frac{1}{j\omega C_{a,fv}}\left(2R_{a,bh}+2R_{a,g}+j\omega M_{a,m}+\frac{1}{j\omega C_{a,m}}+\frac{1}{j\omega C_{a,bv}}\right)}{\frac{1}{j\omega C_{a,fv}}+2R_{a,bh}+2R_{a,g}+j\omega M_{a,m}+\frac{1}{j\omega C_{a,m}}+\frac{1}{j\omega C_{a,bv}}}$$
where(4)$$R_{a,h}=\frac{\rho_{0}}{\pi a_{a,h}^{2}}\sqrt{2\omega \mu }\left(\frac{L_{a,h}}{a_{a,h}}+2\right)$$(5)$$R_{a,bh}=\frac{72\mu h_{bp}}{\pi n_{h}r_{h}^{4}}$$(6)$$R_{a,g}=\frac{12\mu }{\pi n_{h}x_{0}^{3}}B\left(A_{r}\right)$$(7)$$B\left(A_{r}\right)=\frac{1}{4}\ln\left(A_{r}\right)-\frac{3}{8}+\frac{1}{2}A_{r}-\frac{1}{8}A_{R}^{2}$$(8)$$M_{a,h}=\left(L_{a,h}+1.7a_{a,h}\right)\frac{\rho_{0}}{\pi a_{a,h}^{2}}$$(9)$$C_{a,fv/bv}=\frac{V_{a,fv/bv}}{\rho_{0}c_{0}^{2}}$$(10)$$M_{a,m}=\frac{M_{m,m}}{S^{2}}$$(11)$$C_{a,m}=\frac{\pi a_{a}^{6}\left(1-\nu^{2}\right)}{16Eh_{m}^{3}}$$
where *ρ*_0_ is the density of air, *c*_0_ is the speed of sound in air, *a_a,h_* is the radius of the vent, *μ* is the kinematic viscosity of air, *L_a,h_* is the length of the vent, *h_bp_* is the thickness of the backplate, *n_h_* is the number of backplate holes, *r_h_* is the radius of the backplate holes, *x_0_* is the initial distance between the membrane and the backplate. *A_r_* is the perforation ratio of the backplate; *V_a,fv_* and *V_a,bv_* are the volumes of the front and rear cavities, respectively; *S* is the effective area of the membrane; and *M_m,m_* is the effective mass of the membrane.

According to Figure 3, the sound pressure striking the membrane is(12)$$p_{d}=\frac{p_{in}}{Z_{a,tot}}\cdot \frac{1}{j\omega C_{a,m}}\cdot \frac{\frac{1}{j\omega C_{a,fv}}}{\frac{1}{j\omega C_{a,fv}}+2R_{a,bh}+2R_{b,g}+j\omega M_{a,m}+\frac{1}{j\omega C_{a,m}}+\frac{1}{j\omega C_{a,bv}}}$$

To determine the voltage output from the membrane’s average deflection, the conversion coefficient *n* must be analyzed. According to Equation (2), the voltage output for one capacitor is(13)$$V_{out1}=V_{B}\frac{x^{\prime }}{x_{0}}=-\frac{p_{d}a_{a}^{4}}{120D}\frac{V_{B}}{x_{0}}$$
where *V_B_* is the bias voltage.

The output is the difference between capacitors C1 and C2. The electro-acoustic conversion coefficient *n* is given by(14)$$n=2\frac{V_{out1}}{p_{d}}=-\frac{a_{a}^{4}}{60D}\frac{V_{B}}{x_{0}}$$

Therefore, the total voltage output is(15)$$V_{out}=np_{d}=-\frac{V_{B}}{x_{0}}\cdot \frac{a^{4}}{60D}\cdot \frac{p_{in}}{Z_{a,tot}}\cdot \frac{1}{j\omega C_{a,m}}\cdot \frac{\frac{1}{j\omega C_{a,fv}}}{\frac{1}{j\omega C_{a,fv}}+2R_{a,bh}+2R_{b,g}+j\omega M_{a,m}+\frac{1}{j\omega C_{a,m}}+\frac{1}{j\omega C_{a,bv}}}$$

The frequency response is(16)$$S_{sen}=20\log_{10}(\left|V_{out}\right|/p_{in})$$

The parameters used in the aforementioned LPM model are listed in Table 1.

## 3. Finite Element Simulation and Experiment

Finite element simulations of the frequency response curves were conducted for the dual-backplate MEMS microphone in this section, and the frequency response of corresponding samples was experimentally tested. Simulations were conducted using the commercial software COMSOL Multiphysics (version number 6.3). Testing was performed in a fully anechoic chamber at the National Institute of Metrology, China (Changping Facility).

### 3.1. Finite Element Simulation

Finite element simulations employed four coupled physical fields: thermoviscous acoustics, membrane, electrostatics, and electrical circuits. Thermoviscous acoustics modeled the acoustic behavior (including viscous losses) within the microphone’s air domain. Membrane mechanics described the mechanical response of the microphone diaphragm to external sound pressure. Electrostatics computed the output voltage signal between the diaphragm and backplates. It should be noted that the electrostatic field must be coupled with moving mesh settings to correctly compute the output voltage signal between electrodes. The moving mesh domain is defined as the air region between the two backplates (including perforations in the backplates), with fixed boundaries assigned to the surfaces of the backplates adjacent to the diaphragm. The electrical circuit provided bias voltages (40 V) across the diaphragm and dual backplates.

Figure 4 shows the geometric configuration used in the model, with the package and PCB modeled as rigid walls. The model includes the air domain (vent, front cavity, and rear cavity). Sound enters through the vent, travels through the front cavity to the top backplate, and then through the backplate holes to the membrane, causing a capacitance change and a corresponding voltage shift. As the membrane moves towards one backplate and away from the other, the voltage across the capacitors is inverted. The differential signal between the two outputs represents the total output of the dual-backplate structure. In the simulation, a fixed constraint was applied to the outer edge of the diaphragm to replicate the boundary conditions imposed by the support structure. A 1 Pa sound pressure excitation was applied to the exterior surface of the vent hole, and the output voltage signal between the diaphragm and backplate was computed.

The sensitivity in the simulation is(17)$$S_{sen}=20\log_{10}\left(\frac{\left|V_{out1}-V_{out2}\right|}{p_{in}}\right)$$
where *V_out_*_1_ and *V_out_*_2_ are the voltage outputs of the top and bottom capacitors, respectively.

The geometric and material parameters for the finite element simulation are detailed in Table 2 and Table 3, respectively. The primary structural dimensions and material parameters described above were provided by engineers at Gettop Acoustics Co., Ltd (Weifang, China). These dimensional parameters were validated through industrial-grade computed tomography (CT) scanning with 2 μm resolution.

### 3.2. Experiment

To verify the accuracy of the LPM and FEM models discussed above, this paper conducted a sensitivity test on a dual-backplate MEMS microphone sample. The device under test (DUT) was a commercial dual-backplate MEMS microphone, model MD-HRA371-H10-1, manufactured by Gettop Acoustics Co., Ltd (Weifang, China). Key specifications provided by the manufacturer include a sensitivity of −38 dBV/Pa, a signal-to-noise ratio of 66 dBA, and an operating voltage of 1.8 V. The experimental setup is illustrated in Figure 5, where the Device Under Test (DUT) microphone is aligned with a calibrated 1/2-inch reference microphone, maintaining a 50 cm distance from the speaker. The entire testing process was carried out in an anechoic chamber, and the arrangement of the DUT microphone is shown in Figure 6. The test signal input to the speaker is a 1/3 octave band sine wave sweep. The 1/2-inch reference microphone (Type 4189, Bruel & Kjaer, Nærum, Denmark) was calibrated using a pistonphone (Type 4228, Bruel & Kjaer, Nærum, Denmark) prior to measurement. The speaker was driven to produce a sound pressure level (SPL) of 94 dB (1 Pa) at the microphone position for the 1/3 octave band sine sweep.

The frequency response curves of the MEMS microphone sample are shown in Figure 7. Sensitivity tests were conducted on three samples of the commercial dual-backplate MEMS microphone (MD-HRA371-H10-1, Gettop Acoustics Co., Ltd, Weifang, China) to ensure reproducibility. The results showed good consistency, and the curve presented in Figure 7 represents the average sensitivity of the tested devices. In the range from 200 Hz to 20 kHz, the numerical results from LPM and FEM are in good agreement with experimental results. However, below 200 Hz, the test sensitivity of the sample drops sharply. The LPM model did not predict this phenomenon, while the FEM model only shows a sharp drop in sensitivity below 100 Hz. This roll-off behaves as a first-order high-pass filter, with a cutoff frequency f_c_ (f_c_ = 1/(2πR_out_C_mic_)) determined by the preamplifier’s output impedance R_out_ (2 kΩ) and the microphone’s capacitance C_mic_ (3pF). For typical values of R_out_ and C_mic_ in this circuit, f_c_ falls within the 100–200 Hz range, explaining the observed phenomenon. The Finite Element Method (FEM) can simulate this phenomenon through multiphysics coupling settings, whereas the Lumped Parameter Model (LPM) neglects this aspect. However, increasing the preamplifier output impedance can mitigate the low-frequency roll-off in the frequency response. Reference [12] also showed that creating a vent on the package cover to equalize the pressure difference between the front and rear cavities can flatten the frequency response curve below 100 Hz. The reduction in sensitivity predicted by FEM has an error of about 2 dB compared to the measurement. It is evident that, although FEM has lower computational efficiency than LPM, its computational accuracy is higher. Overall, the measured curve shows good consistency with the curve from the numerical calculation, indicating the accuracy of the LPM model and the FEM model. The LPM model and FEM model used above can both be used to parameter analysis of MEMS microphones. However, the FEM model is more suitable for analyzing small-scale structures due to its higher computational accuracy compared to the LPM model. The following text will use an FEM model to analyze the key parameters that impact the sensitivity of dual-backplate MEMS microphones.

### 3.3. Sensitivity Determinant Characterization

Numerical simulations were systematically conducted to identify critical design parameters influencing microphone sensitivity through geometric and material analyses. While prior research by Chengpu Sun [15] established correlations between sensitivity and bias voltage, vent diameter, and front cavity volume (governed by cavity length), these parameters are excluded from the present investigation.

The study focuses on three underexplored sensitivity determinants: Backplate perforation density, Membrane tension, and Electrode gap spacing.

Notably, the silicon-based diaphragm material exhibits temperature-invariant mechanical properties (<2% Young’s modulus variation across −40~100 °C) as demonstrated in reference [16], thereby eliminating thermal analysis requirements.

#### 3.3.1. Backplate Perforation Density

A computational investigation was conducted to characterize the influence of backplate perforations on the frequency response of a dual-backplate MEMS microphone.

Systematic finite element analyses evaluated two geometric parameters: hole diameter and number. For the analysis of hole diameter, the number of holes was set to 41, and the hole positions were symmetrically distributed, as shown in Figure 8 (Left). For the analysis of the number of holes, the hole diameter was fixed at 90 µm, and the hole positions were also symmetrically distributed. Simulations were conducted for three configurations with 41, 25, and 13 holes, respectively, as illustrated in Figure 8. The holes are arranged in a concentric circular pattern with uniform angular spacing. The hole edges in the model are considered sharp, as the focus was on the global effect of perforation density rather than localized edge effects. Frequency response curves are shown in Figure 9. For the parametric studies in this section, the baseline hole radius used in simulations is 45 μm, unless otherwise specified for the specific study on hole diameter.

As shown in Figure 9a, the diameter of the holes on the backplate significantly affects the frequency response near f0. A reduction in hole diameter leads to a notable decrease in the amplitude of the frequency response around near f0 and its adjacent frequencies, accompanied by an expansion of the affected frequency range. Furthermore, a pronounced dip emerges before near f0, which becomes more evident with smaller hole diameters. This phenomenon arises because smaller holes intensify the squeeze-film damping and acoustic resistance within the holes, inducing nonlinear damping effects.

Additionally, a reduction in the hole diameter mitigates the roll-off phenomenon in the frequency response curve within the low-frequency range (<100 Hz). Reference [12] reported that the phenomenon of frequency response drop at low frequencies can be mitigated by creating small holes on the wall surface of the packaging shell.

Similar trends are observed in Figure 9b. When the hole diameter remains constant, a reduction in the number of holes also causes amplitude attenuation in the frequency response near *f*0, along with anti-resonance-like dips. This occurs because fewer holes similarly amplify squeeze-film damping, leading to nonlinear damping effects. Similarly, a decrease in the number of holes also reduces the extent of the roll-off in the frequency response curve below 100 Hz.

#### 3.3.2. Membrane Tension

The frequency response of the capacitive microphone is heavily influenced by the membrane tension, a parameter determined by the membrane’s thickness and fabrication process. To gain deeper insights into this relationship, we performed a series of parametric analyses by varying the tension of the membrane. The sensitivity of the dual-backplate configuration was then evaluated at a frequency of 1 kHz. The outcomes of these analyses are illustrated in Figure 10.

Figure 10 demonstrates that the membrane tension significantly influences the sensitivity. As the membrane tension increases, the sensitivity exhibits a nonlinear decreasing trend. Extrapolating from the curve trends, the sensitivity is expected to stabilize when the membrane tension exceeds 7000 N/m and continues to rise, whereas it increases rapidly when the tension falls below 1500 N/m and further decreases.

Notably, the membrane tension cannot be indefinitely increased or reduced. Excessively high tension may induce mechanical failure of the membrane, while excessively low tension risks pull-in instability between the membrane and the backplate. Both scenarios would lead to microphone failure.

#### 3.3.3. Electrode Gap Spacing

The subject of this study is a dual backplate microphone, which has a diaphragm–backplate structure different from that of the dual-diaphragm microphone. In theory, the effect of Electrode gap spacing on both microphone structures should follow the same pattern. To further confirm this hypothesis, the study re-conducted the simulation analysis, and the results are shown in Figure 11.

From Figure 11a, it can be observed that the electrode gap spacing has a significant impact on the overall amplitude of the frequency response curve. With smaller electrode gap spacing, the overall amplitude of the frequency response is higher, and within the frequency range below 100 Hz, the larger the gap, the greater the rate of decrease in the curve. In the frequency range where the frequency response curve is relatively stable (100 Hz to 20 kHz), the electrode gap spacing almost exclusively affects the amplitude of the curve.

The simulation results, as shown in Figure 11b, confirm that the influence of electrode gap spacing on the dual-backplate microphone follows the same trend as observed in dual-diaphragm architectures [15]. Specifically, the sensitivity varies nonlinearly with the electrode gap spacing, with smaller gaps resulting in higher sensitivity. This consistency underscores the universal applicability of this design principle across different capacitive MEMS transducer topologies.

## 4. Conclusions

This study systematically evaluates the sensitivity of dual-backplate capacitive MEMS microphones through multi-physics modeling and experimental characterization. Key findings include:

Backplate Perforation: Reducing the hole diameter or count intensifies squeeze-film damping, causing amplitude attenuation near f0 and anti-resonance dips. However, it mitigates low-frequency roll-off (<100 Hz), aligning with prior packaging ventilation strategies.

Membrane Tension: Sensitivity decreases nonlinearly with increasing tension, stabilizing above 7000 N/m. Excessive tension risks mechanical failure, while insufficient tension induces pull-in instability, necessitating balanced design.

Electrode Gap: Smaller gaps improve sensitivity but face pull-in voltage constraints, echoing trends in dual-diaphragm architectures.

The FEM model outperforms LPM in capturing low-frequency anomalies (<200 Hz), albeit with higher computational costs. Experimental validation confirms the models’ accuracy (≤2 dB error), supporting their use in parameter optimization. Future work should explore thermal effects, advanced damping mitigation, and multi-objective design frameworks to further enhance performance. These insights advance the development of high-sensitivity MEMS microphones for applications in IoT, biomedical devices, and acoustic sensing.

## Figures and Tables

**Figure 1 micromachines-16-01154-f001:**
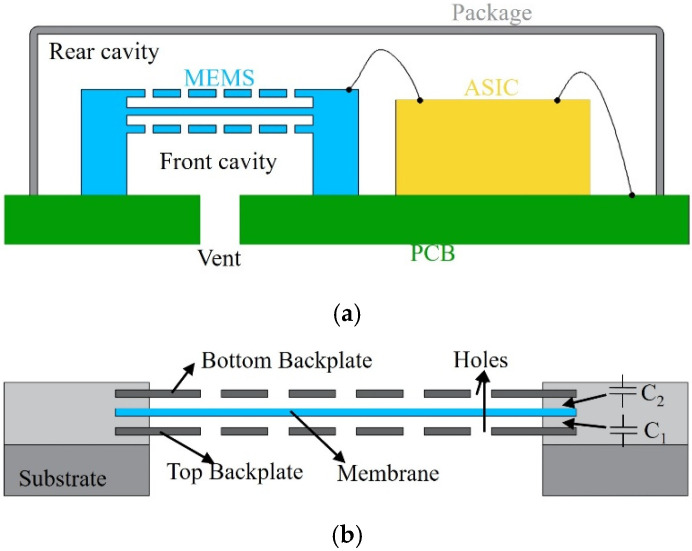
Dual-backplate capacitive MEMS microphone. (**a**) Microphone chip. (**b**) Dual-backplate MEMS.

**Figure 2 micromachines-16-01154-f002:**
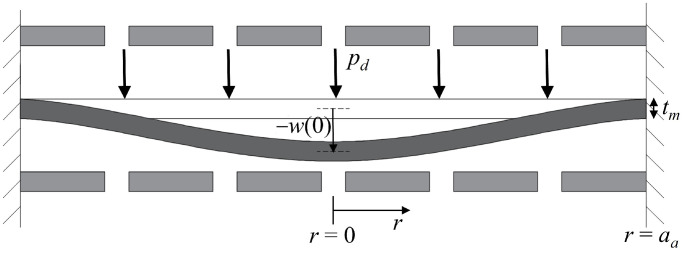
Cross-section of the idealized circular dual-backplate with a small deflection.

**Figure 3 micromachines-16-01154-f003:**
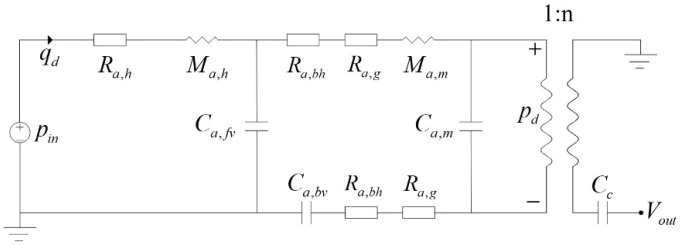
Lumped element model of the dual-backplate microphone.

**Figure 4 micromachines-16-01154-f004:**
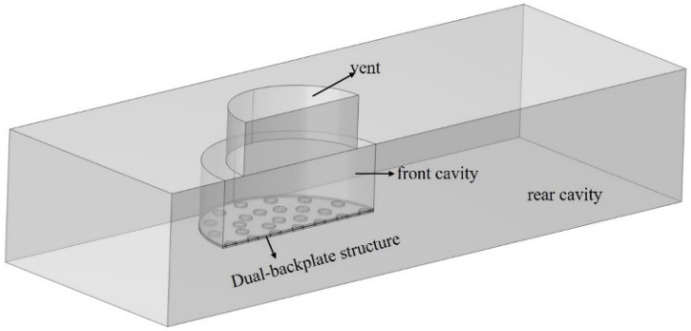
Simulation geometry of the dual backplates MEMS microphone.

**Figure 5 micromachines-16-01154-f005:**
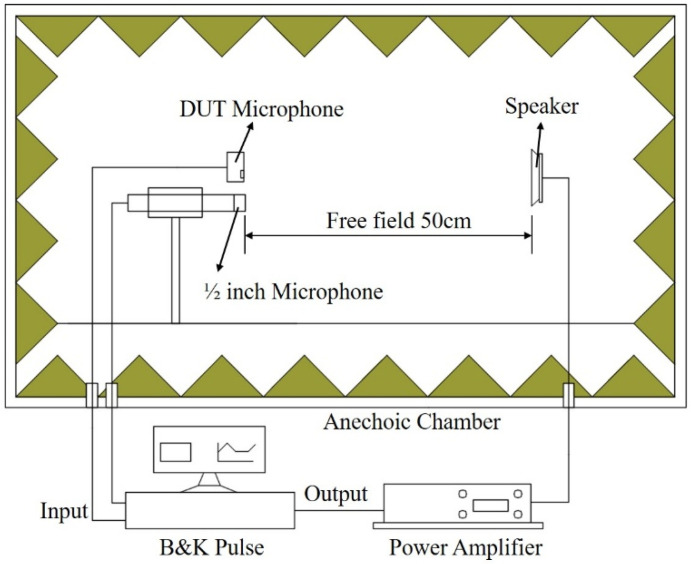
Schematic Diagram of the Experimental Test Device.

**Figure 6 micromachines-16-01154-f006:**
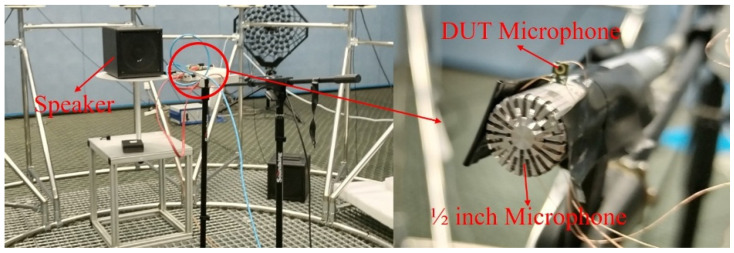
Experiment setup: DUT microphone in anechoic chamber. The 1/2 inch reference microphone is a B&K Type 4189.

**Figure 7 micromachines-16-01154-f007:**
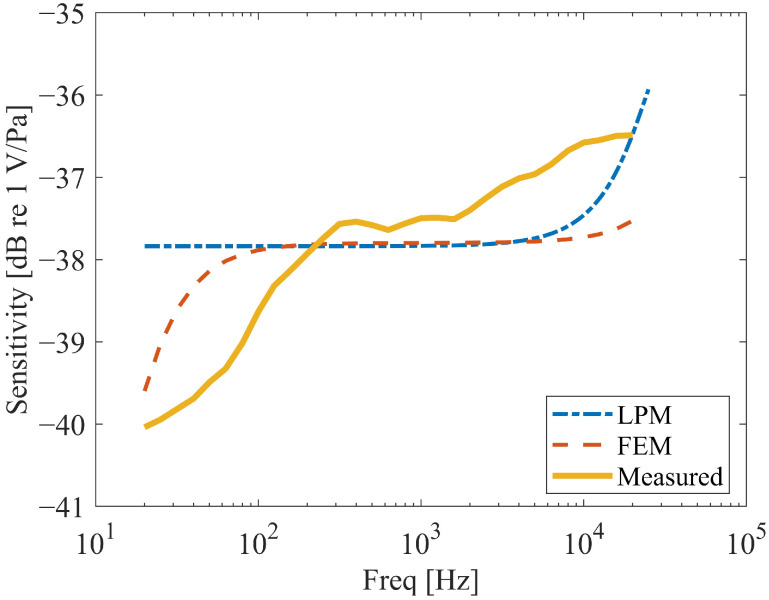
Frequency response of DUT microphone.

**Figure 8 micromachines-16-01154-f008:**
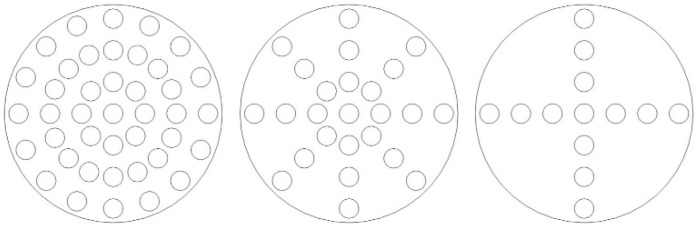
Distribution of Small Hole Positions on the Backplate (From (**left**) to (**right**): 41 holes, 25 holes, 13 holes).

**Figure 9 micromachines-16-01154-f009:**
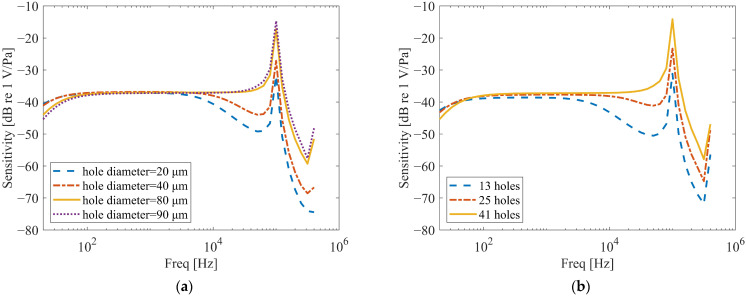
Frequency response in different Backplate perforation densities. (**a**) The effect of the hole’s diameter on frequency response (hole number is 41). (**b**) The effect of the number on the frequency response (hole diameter is 90 µm).

**Figure 10 micromachines-16-01154-f010:**
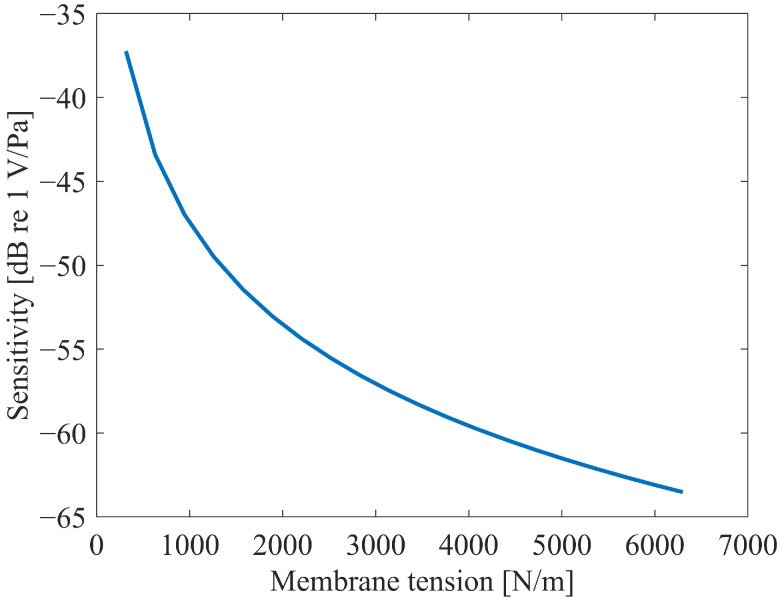
The effect of the membrane tension on sensitivity at 1 kHz.

**Figure 11 micromachines-16-01154-f011:**
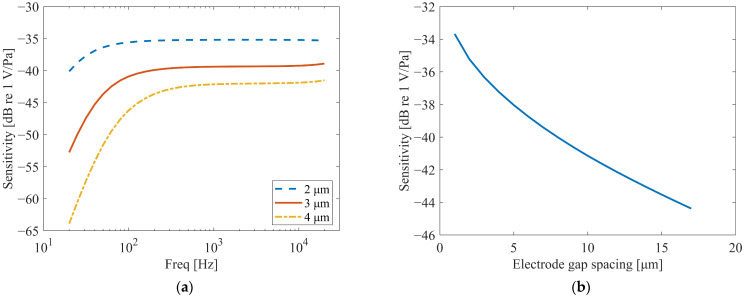
Effect of the Electrode gap spacing. (**a**) The effect on the frequency response. (**b**) The effect on sensitivity at 1 kHz.

**Table 1 micromachines-16-01154-t001:** Acoustic lumped element values of the microphone.

Symbol	Description	Value	Unit
*R_a,h_*	Acoustic resistance of the vent	0.6 × 10^6^~19 × 10^6^ (20 Hz–20 kHz)	N·s/m^5^
*M_a,h_*	Acoustic mass of the vent	2.87 × 10^3^	kg/m4
*C_a,fv_*	Compliance of the front cavity	1.28 × 10^−15^	m^5^/N
*R_a,bh_*	Acoustic resistance of the backplate holes	2.79 × 10^6^	N·s/m^5^
*R_a,g_*	squeeze-film damping	2.48 × 10^9^	N·s/m^5^
*M_a,m_*	Acoustic mass of the membrane	1.36 × 10^4^	kg/m^4^
*C_a,m_*	Compliance of the membrane	1.02 × 10^−15^	m^5^/N
*C_a,bv_*	Compliance of the rear cavity	2.75 × 10^−14^	m^5^/N
*p_in_*	Sound pressure input	1.00	Pa
*D*	Flexural rigidity of the membrane	4.55 × 10^−7^	N·m
*n*	Electro-acoustic conversion coefficient	1.37 × 10^−2^	

**Table 2 micromachines-16-01154-t002:** Microphone Geometric Parameters.

Symbol	Description	Value	Unit
*a_a_*	Radius of membrane	1.0	mm
*h_m_*	Membrane thickness	2.0	µm
*h_bp_*	Backplate thickness	2.0	µm
*x* _0_	Electrode gap spacing	2.4	µm
*d_h_*	Diameter of the vent	0.66	mm
*L_a,h_*	Length of vent	0.25	mm
*r_fv_*	Diameter of the front cavity	0.88	mm
*h_fv_*	Length of the front cavity	0.3	mm
*r_h_*	Radius of the backplate holes	45	µm
*a_v_*	Length of the rear cavity	3.0	mm
*b_v_*	Width of the rear cavity	2.2	mm
*h_v_*	Height of the rear cavity	0.62	mm

**Table 3 micromachines-16-01154-t003:** Material Properties Of Membranes.

Symbol	Description	Membrane	Backplate	Unit
*ρ*	Density	2300	8300	kg/m^3^
E	Young’s modulus	168	221	GPa
*ν*	Poisson’s ratio	0.2	0.3	

## Data Availability

The original contributions presented in the study are included in the article, further inquiries can be directed to the corresponding author.

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
