# Peer review of "Analysis of Key Factors Affecting the Sensitivity of Dual-Backplate Capacitive MEMS Microphones"

_micromachines, 2025, doi:10.3390/mi16101154_

Round 1
Reviewer 1 Report
Comments and Suggestions for Authors
This paper presents a comprehensive investigation of sensitivity-determining factors in dual-backplate capacitive MEMS microphones through analytical modeling, finite element analysis (FEM), and experimental validation. The study focuses on three critical design parameters: backplate perforation density, membrane tension, and electrode gap spacing.This work provides systematic guidelines for optimizing dual-backplate MEMS microphone designs, balancing sensitivity, stability, and manufacturability. But there are still several issues that need to be addressed:
- Section 2.2 clearly identifies the diaphragm's bending stiffnessD as the core parameter for deriving sensitivity, yet it only addresses the determination of D with a brief statement "confirmed via finite element analysis" and fails to disclose any calculation details.
- In Section 2.3, the double-backplate structure is required to adopt the topology shown in Figure 3, which is simplified into a series structure of "2Rₐ,bh + 2Rₐ,g". This simplification violates the core acoustic path of the double-backplate microphone presented in Reference [15], which consists of two parallel paths (the upper and lower backplates each correspond to an independent acoustic channel).
- The meaning of "n" in Table 1 is unclear, and it can easily be mistaken for the number of transformer turns.
- In Section 3.1, it is clearly stated that "a four-field coupling of thermoviscous acoustics, diaphragm mechanics, electrostatics, and circuitry is adopted". However, the 3D simulation diagram does not exhibit any coupling characteristics.
- In Section 3.2, information regarding the test sample (MD-HRA371H10-1) is missing; supplementary information should be provided.
- Section 3.2 simply attributes the inability of the LPM (Lumped Parameter Model) to predict the sharp drop in sensitivity below 200 Hz to "the neglect of the preamplifier's output impedance". However, it neither explains "why this impedance only affects the frequency band below 200 Hz" nor establishes a quantitative relationship between impedance and low-frequency roll-off (e.g., a functional curve between impedance value and roll-off frequency).
- In Section 3.3.1, it is clearly stated that "when analyzing the effect of hole diameter, the number of holes is set to 41; when analyzing the effect of the number of holes, the hole diameter is fixed at 90 μm". However, in Table 2, "backplate hole radius (rₕ)" is marked as "30" (corresponding to a diameter of 60 μm), which completely conflicts with the "hole diameter of 90 μm" (corresponding to a radius of 45 μm) used in the experiment. This results in a failure to align the "modeling parameters" with the "experimental parameters".
- When investigating the effect of backplate perforation density in Section 3.3, only the "symmetric distribution of hole positions" is stated. However, it fails to specify the hole arrangement pattern (e.g., whether it is a concentric circle distribution, or whether the hole spacing is uniform) and whether the hole edges are rounded (which affects airflow damping).
- The authors claim to focus on "parameters that have not been fully studied". However, the conclusion regarding electrode gap in Section 3.3.3 ("a smaller gap improves sensitivity but is limited by the pull-in voltage") is completely consistent with that in Reference [16] (for the dual-diaphragm structure).
- When analyzing three key parameters that affect sensitivity—backplate perforation density, diaphragm tension, and electrode gap—a special case below 200 Hz is mentioned, but no detailed explanation is provided.
Reviewer 2 Report
Comments and Suggestions for Authors
The paper presents a combined analytical, FEM, and experimental study on how backplate perforation density, membrane tension, and electrode gap spacing affect the sensitivity of dual-backplate capacitive MEMS microphones. A LPM and COMSOL FEM are compared against measurements in an anechoic chamber. I would recommend “major revision” before acceptance. Please find below my comments.
1. Only one commercial sample (MD-HRA371-H10-1) is mentioned. Indicate how many devices were tested and show error bars if multiple runs were performed.
2. Please explain calibration of the 1/2-inch reference microphone and the SPL used for excitation.
3. The text attributes the roll-off to preamplifier output impedance but does not specify the actual impedance values used in the test vs. the model. Including these would strengthen the argument.
4. The heading has a typo (“3.3.3 Electrode gap spacin”).
5. Add axis labels with units and legends (e.g., “Sensitivity [dB re 1 V/Pa] vs Frequency [Hz]”) so readers can interpret curves without referring to the text.
6. Figures 5 and 6 show setups but no scale bars or description of the reference microphone model—add these details.
7. Several parameters are defined in the text but not in Table 1, or vice versa (e.g., BBB, ArA_rAr, x0x_0x0). Provide a complete symbol table to aid reproducibility.
Round 2
Reviewer 1 Report
Comments and Suggestions for Authors
I would like to thank the authors for their thorough and careful revision of the manuscript. They have addressed all my previous comments and suggestions point by point, and the revisions have significantly improved the clarity and quality of the work. The manuscript is now acceptable for publication in its present form.
Reviewer 2 Report
Comments and Suggestions for Authors
All my comments have been satisfactorily addressed. The manuscript is suitable for publication without further review.